# An Augustinian Meditation on the *Saeculum*

**Matthew Drever** 

Department of Philosophy and Religion, The University of Tulsa, Tulsa, OK 74104, USA;
matthew-drever@utulsa.edu

**Abstract:** Augustine's account of the *saeculum* brings together his theological anthropology, ecclesiology, and Christology in establishing the church as the body of Christ—as the place wherein the restless and sinful soul finds renewal through the grace of the incarnation. Such a model of the *saeculum* pushes back against various contemporary appropriations of Augustine, including those of Milbank, Markus, and Taylor.

**Keywords:** *saeculum*; secular; Christ; church; body of Christ; the Psalms; politics; soul; ethics; suffering

---

> "What occurs in the psalm as a whole occurs in its particular pieces and its individual syllables. The same is true of a longer action in which perhaps that psalm is a part. It is also valid of the entire life of an individual person, where all actions are parts of a whole, and of the total history of the sons of men [*hoc in toto saeculo filiorum hominum*] where all human lives are but parts". (*Conf.* 11.28.38)

## 1. The *Saeculum* and the Secular

In the concluding paragraphs of Augustine's well-known account of time in *Confessions* 11, he draws a comparison between human life, history, and the recitation of a psalm. Ostensibly, he is offering a parts/whole analogy, comparing the way in which discrete words unite in the overall meaning of a psalm and the way in which discrete moments in human life find meaning and order within the overall life and history of humanity. This comparison frames Augustine's final prayer in book 11 for self-unity amidst the fragmentation and dispersion he experiences within time and history—within the *saeculum*. Readers of Augustine know, however, that his use of the Psalms serves as more than an analogy to understand unity within a reality fraught with division. We get a glimpse of this in *Confessions* 11 in the way that he uses various psalms to structure his concluding prayer.[1] This intimates that, for Augustine, the Psalms are a medium through which human words of praise (speech acts) conform our existence and identity to the divine Word. In this way, Augustine's mediations on and exegesis of the Psalms open into a hermeneutic of performative soteriology that seeks self-unity and unity with God through the recitation of the Psalms.

Taking my cue from *Confessions* 11.28.38, I will offer an Augustinian meditation on human temporality and history—the *saeculum*—as it unfolds within Augustine's search for unity and wholeness. His interpretation of the Psalms is important but so is the way that he brings together human life and history—the individual and the community—in his account of the *saeculum*. In developing this account, we should begin with Augustine's use of the Latin term '*saeculum*', which is translated in various ways, including to mean 'history', 'age', 'time', and 'world'. There is no fixed or easy way to translate the term, and we must be cautious against translating it as an abstract, ideological category. The main danger

---

1. *Conf.* 11.29.39–11.30.40. This echoes a wider strategy in *Confessions* where Augustine often draws on the Psalms to structure and develop his autobiography.

is the etymological proximity between *saeculum* and the English 'secular' and 'secularism'. There is no straightforward translation of Augustine's *saeculum* into either of these English words, especially when such words refer to a philosophical (ideological) framework in which divine governance over the natural and human world is either bracketed or denied. Here, we are well advised to heed Joseph Rivera's warning that, for Augustine, the *saeculum* has a dynamic quality that "in principle cannot achieve closure or the static essentialism of a neutral public space".[2] Paul Griffiths further nuances this point, arguing that there are at least three distinct meanings in the contemporary notion of the 'secular' and that each meaning connects in a different manner with Augustine's use of *saeculum*.[3] Augustine's use of *saeculum* connects most strongly to a contemporary notion of 'secular' that refers to a set of norms, laws, or institutions that is not grounded in or does not require membership within a specific religion and, thus, can be adopted by religious and nonreligious groups alike, albeit for different motives or reasons. For Augustine, such a model of the secular is possible, because society and politics is comprised of Christian and non-Christian groups, and these groups are mixed together in a way that we cannot separate. Augustine also uses *saeculum* on a limited basis to designate the absence of religious institutions within the state. He uses *saeculum* in this way, however, in a historical, not a teleological, manner, designating not the future direction of society but, rather, ancient states that predate Christianity and Judaism. Such states are secular, because, for Augustine, all bona fide forms of religious institutions derive from Judaism and Christianity. Augustine does not, however, employ *saeculum* to designate a public space that is neutral to or absent of religion or God's presence. Such a notion of the secular would be foreign and antithetical to Augustine, because he connects the normative value and goodness of all human laws back to God.

In further detailing Augustine's notion of the *saeculum*, Griffiths argues that he employs the term in two main ways, both of which underscore further the complexities of rendering *saeculum* within the range of meanings often associated with the modern idea of the 'secular'.[4] First, *saeculum*, as in "*hoc saeculum*", can refer to this time, designating the time and history that constitutes creation, especially the time between the fall in the Garden of Eden and the second coming of Christ. Augustine sometimes further divides this time into six periods.[5] While much of this time is spent in rebellion against God, it is far from devoid of divine governance. Indeed, the times are awash with God's presence, but it is a presence characterized by mystery and ambiguity. This leads into the second, eschatological use of the term, as in "*saeculum futurum*", which designates the eternal age whose coming marks the end of the first age (creation) and is inaugurated by the second coming of Christ. God's governance (predestination) spans both ages and works within divine rhythms that are impenetrable to human reason.[6] However, it is not simply God that is cloaked in mystery. We ourselves live in this age (*hoc saeculum*) and are moving toward the coming age (*saeculum futurum*), and our lives, governed as they are by God, are also shrouded in mystery and ambiguity.[7] To borrow from Augustine's metaphor of the two cities—the earthly and the heavenly city—God may know the place to which each belongs, but this is an eschatological judgment to which we are not privy. We live here and now in the great *corpus permixtum* surrounded by saints and sinners, uncertain of our own or our neighbor's citizenship.[8] In this, Wetzel and Rivera are both right that the sense of the 'secular' most at home in Augustine's writing is one that beckons not toward an ethical and religious neutrality within a political and social

---

[2]　(Rivera 2018, p. 63).

[3]　(Griffiths 2012).

[4]　(Griffiths 2012, pp. 33–34).

[5]　*Gn. adv. Man*. 1.23.35–1.24.41.

[6]　*praed. sanct*. 6.11, 8.16, 18.36, 20.40; *persev*. 1.1, 7.15, 8.18, 9.21, 11.25, 11.27, 13.33.

[7]　*Conf*. 10.29.40–10.30.42; *trin*. 1.13.28, 1.13.30–31, 8.4.6. Michel Barnes emphasizes that, for Augustine, the vision of God is eschatological and that, here and now, God is accessible only through faith, ever veiled in mystery and unable to be known directly (Barnes 2003).

[8]　*doc. Chr*. 3.32; *civ. Dei* 18.49; *persev*. 13.33.

space but, rather, toward the mixed nature of this age and the mystery and ambiguity this age bears as a consequence.[9]

Augustine's account of the *saeculum*, then, should give us pause when we turn to contemporary debates on history, temporality, and the secular. His relation to such contemporary debates is often articulated vis-à-vis his politics and his account of the two cities. The interpretations here are wide ranging and run the interpretative gamut. On one end, Markus charts a path through such waters, arguing that the mystery and ambiguity built into Augustine's two cities metaphor lead inadvertently into contemporary liberal secularism.[10] On the other end, Milbank charts an alternate path, arguing that Augustine's two cities model challenges the contemporary, liberal sacred–secular distinction and leads instead to a close alignment between the sacred and secular.[11] Both accounts have encountered significant criticism, and my purpose is not to revisit substantively either of them. Rather, they will serve as occasions, or channel markers, if you will, to navigate an Augustinian course through contemporary debates on the nature of the 'secular'.

Here, I am prone to agree, again, with Wetzel's assessment that the language of the two cities is something of a misnomer. It serves a rhetorical function for Augustine and corresponds to an eschatological reality, but this is not a reality that is found here and now. In our current times, there is really one city populated with the redeemed and damned.[12] Wetzel's rethinking of the two cities metaphor invites further rethinking and, I would suggest, even a change in metaphors. Here, I would like to chart Augustine's account of the *saeculum* not through the two cities and what goes under the contemporary heading of politics but, rather, through his ecclesiology and his model of the body of Christ. Augustine often develops the body of Christ metaphor in connection with the two cities metaphor.[13] For example, in his explication of Psalm 61, he shifts between the two sets of metaphors, aligning the heavenly city with the body of Christ.[14] Similar to Augustine's account of the two cities, the body of Christ is foremost a mystical, eschatological metaphor, connected to the perfection of the Church at the resurrection and the unity between Christians and God.

The critic might retort that, in shifting metaphors, I have drifted into the territory of Milbank, conflating the body politic with the body of Christ—the sacred with the secular. Left unchecked, this might even reposition Augustine within the Eusebian imperialism he comes to eschew. My point, however, is not to advocate for such a position but, rather, to seek an alternate vantage point on Augustine's understanding of the *saeculum*, one that intentionally begins with the second rather than the first Adam—with grace rather than sin.[15] I have a few reasons for doing this. First, there is no shortage in Augustine's writing on either the topic of sin or grace, but contemporary accounts of Augustinian politics, especially those that favor liberal secular models, often overstate the former at the expense of the latter.[16] This is especially true when such contemporary accounts encroach on Augustine's doctrine of predestination. As I hope to show, there is something to be gained by beginning with the predestination of Christ and the way that we participate in this through our inclusion within Christ's body rather than with the predestination of Adam and our inclusion within his sin.[17]

---

9   (Wetzel 2004; Rivera 2018, pp. 65–67).
10  (Markus 1970, pp. 55, 64, 69–71, 83–84, 101, 104, 125–26, 133, 151).
11  (Milbank 2006, pp. 382–442).
12  (Wetzel 2004, pp. 275–76).
13  (Markus 1970, p. 117).
14  *En. Ps.* 61.4–10.
15  Here, I again agree with Rivera who argues that, for Augustine, Adam is an archetypal figure who represents the basic existential possibilities—under the conditions of sin and grace—for humans to exist within the *saeculum* (Rivera 2018, pp. 68–70).
16  Eric Gregory argues for the need for greater balance, pointing to the connection between approaches, similar to that of Markus, which emphasize Augustinian accounts of sin and a premature desacralization of politics (Gregory 2001).
17  I acknowledge that this moves against Augustine's own rhetorical flow. For example, in *praed. sanct.*, he spends much of the opening of the text arguing about our relation to the sins of Adam before moving on in 15.30ff to describe our inclusion within the predestining grace of the incarnation.

Second, such an approach may help uncover certain contemporary biases. The two cities metaphor may appear to modern eyes to be a more appropriate basis to analyze questions of history and secularity, because it seems more political and less overtly religious than the body of Christ. In fact, however, both metaphors are eschatological and should come with warnings against facile applications to politics.[18] Shifting metaphors to the body of Christ will, at least for contemporary eyes, highlight the ambiguity of Augustine's interpretation of the *saeculum* and the complexities entailed in applying his eschatological metaphors to political and sociological realities. In doing so, it will avoid a standard contemporary route via Augustine's politics to a conception of the 'secular'—one that is often already decoupled from the sacred. That is, there is a modern tendency, foreign to Augustine, which seeks analyses and delineations of secularity within the framework of politics and takes this as the right place from which to understand the secular on the prior twofold assumption of a distinction between the sacred and secular and that the science of politics offers the best method of articulating the space of secularity.

Third, another advantage in shifting from the two cities to the body of Christ is the way that it connects us back to *Confessions* 11 and to Augustine's Christological exegesis of the Psalms. Through this exegetical method, Augustine seeks to discern the relation between the individual, the Church (body of Christ), and Christ (the head). This will open us to the question of how the individual and the community embody the times—the *saeculum*. We will see that these are not distinct and discontinuous accounts of temporality—of the *saeculum*—but that, rather, there is a coherency and unity in Augustine's Christology in the way that the individual, as the image of God, participates in the body of Christ and so imitates Christ, who is the perfect divine image. This will also invite a further reassessment of Augustine's writing, focused now on his account of human selfhood and its connection with modern and secular models of the human person. Here, again, we will see that scholars have developed a range of interpretations on this issue. On the one hand, Charles Taylor argues that an Augustinian account of the self foreshadows and foregrounds modern notions of selfhood.[19] On the other hand, Michael Gillespie contends that an Augustinian account feeds into a medieval synthesis of human selfhood, whose breakdown during the Reformation leads into an alternate modern notion of selfhood.[20] Here, I will push back against both sets of accounts, seeking rather to chart a path through Augustine's Christology toward an understanding of how, within the *saeculum*, he articulates human selfhood (i.e., the history of the individual) and its connection with wider society (i.e., human history). Such an account complements the work of Joseph Rivera, who argues that the theological anthropology that Augustine develops in *City of God* XV, which is grounded in a Pauline flesh/spirit dichotomy, underlies his later social and political account of the *saeculum* in book XIX.[21]

## 2. The Times I: Human Selfhood

In tracing the roots of modernity and its discontents, Charles Taylor argues that Augustine is the unwitting progenitor of the modern Cartesian subject. According to Taylor, Augustine grounds the search for truth within the radical reflexivity of first-person subjectivity, which leads Augustine to develop a version of the anti-skeptical argument Descartes will later make famous.[22] Other commentators join Taylor and contend that Augustine brings together Neoplatonist and Christian

---

[18]　See Bowlin's critique of both Milbank and Markus (Bowlin 1997).

[19]　C. Taylor (1989, pp. 127–42). Though not in line with his wider project, Taylor's treatment of Augustine has parallels with Löwith's secularization thesis, namely, that modernity builds off the secularization of Medieval Christian concepts (Löwith 1949).

[20]　(Gillespie 2008, pp. 1–18, 131–135, 141–44, 151–61).

[21]　(Rivera 2018, pp. 68–72).

[22]　C. Taylor (1989, pp. 131–33). Taylor defines radical reflexivity as a type of first-person, existential knowledge: it is the immediate awareness that distinguishes my experience of myself from my experience of other people. Gareth Matthews acknowledges differences between the two thinkers but also argues for a close enough relation between them that he can refer to Augustine's view as a "nearly Cartesian philosophy of mind". (Augustine 2002, p. xii).

anthropologies, becoming one of the first Western thinkers to articulate human subjectivity along the modern lines of an inward, autonomous, rational space where human identity is self-constituted through an act of freewill.[23] Michael Gillespie agrees with the underlying presupposition that guides such projects, namely, that philosophical and theological notions of human selfhood, and their ability to address questions of agency and identity, are fundamental to conceptions of modernity and secularity.[24] Gillespie disagrees, however, with the thesis on the continuity between Augustine and modernity, arguing, instead, that modernity emerges as a solution, though one with inherent problems, to the failure of Reformation and Humanist debates to resolve the crises that developed as a result of nominalist critiques of the Medieval (and Augustinian) synthesis on human agency and identity. In Gillespie's model, Augustine lays the groundwork for modernity in the negative rather than the positive, as an accomplice to the failed Medieval system that leads into the so-called crisis of modernity. In developing his account, Gillespie articulates a notion of modern subjectivity similar to the above commentators but also highlights a point important for my analysis, namely, that the temporal dimension is fundamental to the modern articulation of self-identity. The formation of human identity within time, the autonomy over this formation, and so, then, a mastery and control over time (and the times) is basic to modern conceptions of selfhood.[25] Such conceptions lay the groundwork for modern secularism as the right, obligation, or duty of humanity to define and control the times against theological claims of divine prerogative.

Gillespie and Taylor may disagree over Augustine's place within the movement toward modern secularism, though both thinkers grant him the dubious honor as either its progenitor (Taylor) or a party to the collapsed Medieval synthesis that leads to it (Gillespie). I disagree with both assessments of Augustine, though I agree with the more general impulse of each scholar that there is a relation between conceptions of space, place, and time—the *saeculum*—and theological models of human subjectivity. This is in keeping with a basic theme that emerges within contemporary scholarship—Heideggerian phenomenology, Gadamerian hermeneutics, and the later Wittgensteinian linguistic analysis—that human existence is a project of sorts, subject to construction and reconstruction, that relates to the distinctive nature of human temporality and its place—its environment in the deepest, existential sense of the term. The question is as follows: how does Augustine characterize the inward self and what lessons can we draw from its connection to the space, place, and times of the *saeculum*? Qualifying Taylor's thesis, Heidegger argues that Augustine's account of human subjectivity has a more ambiguous legacy vis-à-vis postmodern projects that attempt to reclaim authentic accounts of temporality.[26] On the one hand, Heidegger feeds into Gillespie's model, arguing that Augustine maintains a classical and problematic account of time insofar as he frames human temporality in and through divine eternity. Here, according to Heidegger, authentic finite human existence is lost under the shadow of divine eternity in the human quest to reunite with God. On the other hand, Heidegger thinks that Augustine beckons toward a modern, existential model of human finitude in *Confessions* 11 in his attempt to capture the nature of temporality within the stretching of the soul—Augustine's famed *distentio animi*.[27]

Heidegger's evaluation of Augustine returns us to *Confessions* 11 and Augustine's evaluation of human temporality. We have already seen that Augustine concludes book 11 by comparing individual life and human history (the *saeculum*) with the recitation of a psalm: within each activity, there is a

---

[23] (Farrell 1994, pp. 8–10; M.C. Taylor 1984, pp. 35–40; Cary 2000, pp. 3, 63–64, 115–22).

[24] Gillespie is building on Hans Blumenberg's well-known argument that modernity develops from a fundamental break with the Christian Medieval world—a break characterized by the modern embrace of the historical world vis-à-vis science against the Christian ascetic rejection of the world (Blumenberg 1983).

[25] (Gillespie 2008, p. 2).

[26] Heidegger develops this argument in his early writings. It receives significant attention in (Heidegger 1992, 2004). There are also passing references to it in *Being and Time*: 44 fn1, 171 fn2, 190 fn1, 427 fn3.

[27] There is a variety of secondary scholarship that evaluates Heidegger's assessment of Augustine. See, for example: (Brachtendorf 2007; de Paulo 2006; Fischer and Hattrup 2006; Mendelson 2000).

process of distinction that occurs within the context of an underlying unity. Beyond serving as an analogy on the relation between parts and the whole, Augustine's wider use of the Psalms plays an essential role that is in the background of his concluding petition in *Confessions* 11 "to hear the voice of praise and contemplate your delight".[28] The Psalms offer such a voice, serving a soteriological function and revealing concomitant judgments on what constitutes genuine and whole (redeemed) human personhood.[29]

Augustine's reference to praise in connection with contemplation and his employment of the Psalms to voice this praise belongs to a wider arc in his thought on the nature of human existence and identity and how we find wholeness and perfection in relation to God. Before turning to Augustine's exegesis of the Psalms, I would like to examine further this arc. We can begin with *On the Trinity*, in which Augustine argues that contemplation finds its fulfillment in wisdom, which is also the worship of God.[30] This worship, or piety, emanates into the reformation and perfection of the divine image at the core of human identity in its relation with God.[31] In this context, the worship that constitutes the reformation of the divine image includes, but is not reducible to, the discrete set of activities—the liturgy, prayer, the recitation of Psalms—performed within the historical Church. Here, worship (piety) refers to the more fundamental existential alignment of the self to God within the unity of intelligence and love—an alignment that opens up the possibility of humans imaging the divine.[32] Discrete liturgical activities have a reforming function insofar as they connect with the more fundamental contours of human existence in its openness and orientation toward God.

The connection that Augustine draws between the existential contours of human selfhood and the liturgical practices of the Church traces back to his intertwined doctrines of church and creation. We glimpse the relation between these doctrines in *Confessions* 13 in which Augustine claims that the church exists already in God's creative actions in Genesis 1.[33] This is not the historic Church as such but, rather, the mystical church grounded in God's predestining will—the same will (grace) at the basis of the incarnation and our inclusion in Christ's body (i.e., the mystical church).[34] The discrete liturgical activities performed in the Church derive their soteriological efficacy in part from their connection with the predestined nature of the mystical church. Such efficacy is also connected to the doctrine of creation that Augustine develops in his *Literal Commentary on Genesis*. Here, he argues that intellectual creation (i.e., angels, humans) exists through its self-constituting comportment in loving openness toward God.[35] This comportment is the existential ground of intellectual creation and the reason why worship as a discrete activity within the Church can function also as a basic (re)articulation of the self and a reformation of selfhood in its fundamental dimensions. Here, worship extends to the origin of the *saeculum* as the condition and possibility for human (re)creation. This original worshipful comportment opens the possibility for deformation in sin and is the original condition glimpsed 'now' in *hoc saeculum* in the disquietude of restlessness and in the yearning for peace and wholeness that points toward the *saeculum futurum*.[36]

The inward space of the soul takes its shape within the complexity of such dynamics. Augustine articulates this space in various ways. There are two examples I would like to highlight, one from *Confessions* and the other from *On the Trinity.* We can begin in *Confessions* 10 and Augustine's pondering

---

28　*Conf*. 11.29.39.
29　*Conf*. 11.29.39.
30　*Trin*. 14.1.1.
31　This concept of piety (*pietas*) is widespread in Augustine's writings. For example, see: *Gn. litt*. 3.20.30–3.20.32, 4.21.38–4.35.56; *ench*. 1.2–1.3; *conf*. 5.5.8, 8.1.2; *spir. et litt*. 10.17–12.20.
32　On unity of intelligence and love vis-à-vis God, see: *Conf*. 13.4.5, 13.7.8; *In ep. Io.* 7.1, 7.2, 7.4, 7.7, 7.9; *trin*. 8.4.6, 8.7.10–8.10.14, 9.10.15–9.12.18.
33　*Conf*. 13.12.13–13.20.26.
34　*praed. sanct*. 15.30–15.31.
35　*Gn. litt*. 1.1.2–1.2.3, 1.4.9, 2.8.16–2.8.19, 3.20.30–3.20.32.
36　On the relation between the existential and eschatological dimensions underlying Augustine's account of the *saeculum*, see also: (Rivera 2018, pp. 65–68).

over the inward space (*memoria*) of the human soul—a space wherein he already confronts and locates the human experience of time.[37] Within this space, within memory, Augustine finds himself: "There also I meet myself and recall what I am".[38] He roots memory firmly within the mind (*animus*), maintaining that the mind is the inward space within which his identity is forged and the difficult project of self-understanding unfolds. Here, it is important to highlight both the mystery and ambiguity that accompany this project. Beginning with the mystery that accompanies selfhood, Augustine observes: "This power of memory is great, very great, my God. It is a vast and infinite profundity. Who has plumbed its bottom? This power is that of my mind and is a natural endowment, but I myself cannot grasp the totality of what I am".[39] Here, already, Augustine confronts the profound depths that underlie the rational space of his mind—depths whose mysteries are impenetrable even as they interpenetrate his mind, forming the conditions of possibility for human rationality.[40] In this context, mystery is a project for the mind in its attempt to uncover the hidden contours of inward human space even as the mind is a product of mystery. At this point, Augustine only alludes to the basis of this latter claim, which is connected to the twofold fact that human existence and identity derive most fundamentally from what we are not: human finite existence derives from the infinite power of God, and human finite identity derives from its imaging of the infinite God.[41] Within this inward space of the mind, the project of recollecting (*cogitare*) the self occurs. This is the task of human thought, and here, Augustine moves through an etymology of 'thinking' (*cogitare*) to portray this task as bringing together (*cogendo*) and gathering (*colligenda*) the contents of memory through the act (*ago*) of thought (*cogitare*).[42] This task of recollecting and gathering the contents of memory is an act infused with love insofar as love comports and binds us to the truth of the objects—the self and God—that it seeks.[43] This loving act is rendered ambiguous, not because of the mystery that accompanies the inward space of the mind but, rather, because of the way in which this inward space has become distorted through sin. As a result of sin, temptations reside within and threaten the inward space of the self, complicating its task of self-understanding.[44]

This is the space within which Augustine first locates the human experience of time.[45] It is this space, rendered ambiguous through sin, to which Augustine returns in *Confessions* 11. Indeed, his attempt to locate time within the stretching of the soul's attention (love) and his attempt to understand the effects of temporality within this space as it exists now gives an answer to the riddle of why Augustine's ostensibly neutral, metaphysical account of time gives way to a testimony of the dangers of human temporality and a soteriological search to remedy the destructive effects of time.[46] The mystery of *memoria* accompanies the ambiguity of human temporality living in the times. The temporal space of the mind stretches (distends) itself through its expectation (future), attention (present), and memory (past)[47] but experiences these in a manner infused with the dynamics of sin and grace: memory (creation, fall); attention (conversion, temptation); and expectation (hope, judgment). Together, then, *Confessions* 10 and 11 explore the existential contours of finite human temporality and the mystery and ambiguity that define it. The space and time of the soul, its temporality, and its way of being in the times (*saeculum*) are made possible through, not against, mystery and are rendered difficult and ambiguous as a result of sin.

---

37 *Conf.* 10.10.17.
38 *Conf.* 10.8.14. "Ibi mihi et ipse occurro meque recolo".
39 *Conf.* 10.8.15. "Magna ista vis est memoriae, magna nimis, Deus meus, penetrale amplum et infinitum. Quis ad fundum eius pervenit? Et vis est haec animi mei atque ad meam naturam pertinet, nec ego ipse capio totum, quod sum".
40 *Conf.* 10.8.13, 10.8.15, 10.16.25–10.17.26.
41 *Conf.* 10.5.7.
42 *Conf.* 10.11.18.
43 *Conf.* 10.6.8–10.7.11, 10.23.34.
44 *Conf.* 10.16.25, 10.29.40–10.30.42.
45 *Conf.* 10.8.14.
46 Compare *Conf.* 11.26.33 and 11.29.39.
47 *Conf.* 11.28.37.

A similar story unfolds in *On the Trinity* through Augustine's examination of the divine image. He affirms again the rational nature of the soul in locating the divine image within the mind (*mens*).[48] However, it is a rational space structured through the dynamics of sin and grace, which becomes apparent when Augustine pauses in the midst of his inward, contemplative search for the divine image (books 8–11) in order to locate this search within a Christian soteriological account of sin (book 12) and salvation (book 13). It is only after this that Augustine articulates the perfection of the divine image (book 14). Books 12 and 13 are not a sidebar before the court of reason, as if faith is a distinct and independent exercise, but, rather, the location and delineation of the rational soul within the narrative of sin and grace. The fulcrum of this account is Augustine's claim that Christ represents the path through knowledge (*scientia*) to wisdom (*sapientia*).[49] Augustine's invocation of Christ signals that, as in *Confessions*, his search for understanding recognizes the intractable ambiguities of this search.

This Christological focus finds an unlikely ally, at least for an audience conditioned through Descartes, in the anti-skeptical exercise that Augustine develops in *On the Trinity* 10.[50] It is a rational exercise, but unlike Descartes' later version of it, Augustine's version does not center on epistemic doubt as such or find resolution in the soul's own rational autonomy.[51] Rather, Augustine focuses on an affective-epistemic doubt, wherein the soul's identity, and thus self-knowledge, is deformed through immoral loves and reformed through the proper reconstitution of its relations.[52] This leads to an anti-skeptical exercise with closer affinities to an *imitatio Christi* than to Descartes' meditative path. The soul seeks to separate out true from false conceptions of itself and to know itself in its true nature.[53] The truth, however, is not an object but a subject—namely, Christ[54]—and is grasped in the reformation of the soul's ownmost identity as the divine image through the imitation of Christ, who is the perfect image of God.[55] Here, the anti-skeptical exercise is an activity of intelligence infused, formed, and fulfilled in acts of love and praise. It is for this reason that Augustine intertwines reason and love in his account of how we arrive at self-knowledge[56] and prefaces his anti-skeptical exercise with his concerns about how the failure to understand oneself properly leads to immoral love and sinful distortions within the self.[57] Likewise, it is for this reason that Augustine also argues that the perfection of the divine image is found within wisdom, who is Christ, and so is grasped in the activity of praise.[58]

Augustine's quest in *On the Trinity* for the contemplation of delight concludes with markers of apophasis.[59] The answer to the problem of self-knowledge is found within, not against, the mystery of self and God. This is an answer to the skeptic but an answer where the fundamental problem is not epistemic self-doubt but, rather, the sinful separation of the self from God. This entails an assertion of mystery at both the epistemic and ontological level. Human knowledge and being, even when perfectly reformed and reunited with God, will not image God as God but as the created image of the divine, that is, as an image whose origin is always radically other (*nihil*) than the eternal simplicity of God.[60] Likewise, human existence and identity forms through its relation to God, but we are a finite image of the infinite and so our ownmost identity always also transcends us. This is not just a

---

48  *trin*. 9.12.17, 10.12.19, 14.8.11; *Gn. litt*. 3.20.30.
49  *trin*. 13.19.24, 14.4.6.
50  (Drever 2013, pp. 124–31).
51  Stephen Menn is careful to note the contextual differences between their versions of the argument and so avoids reading Augustine through Descartes (Menn 1998). Gareth B. Matthews offers a careful evaluation of the relation between Augustine and Descartes, though he tends to read Augustine within a modernist trajectory (Matthews 1992).
52  *trin*. 10.5.7–10.6.8.
53  *trin*. 10.8.11–10.10.14.
54  *trin*. 13.19.24.
55  *trin*. 7.3.5, 7.6.12.
56  *trin*. 8.4.6, 8.7.10–8.10.14, 9.10.15–9.12.18.
57  *trin*. 10.5.7.
58  *trin*. 14.1.1.
59  *trin*. 15.7.11–15.7.13.
60  *trin*. 15.16.26.

condition of knowing but of being. It is the condition of possibility for our existence as such, that is, as created, finite beings. Mystery is not the puzzle that stands against reason but, rather, the condition of possibility for finite reason. Finally, this leads Augustine into the concluding prayer of *On the Trinity*, which petitions for a loving union with God conducted through praise and within the *regula fidei* of the Church.[61]

From Augustine's accounts of the soul in *On the Trinity* and *Confessions*, we see a model of human selfhood at odds with the contemporary critiques coming from scholars such as Taylor and Gillespie. The immediate self-awareness that characterizes the inward nature of the soul is not the self-reflection or self-possession that Taylor worries feeds into modern conceptions of selfhood but, rather, a reflexive openness (worship) to God that itself constitutes the soul's created intellectual space. Augustine conceives of this as a rational space, but it is neither a Cartesian space nor a positivist, empirical space of contemporary scientific reasoning. Rather, it is closer to Tillich's ontological concept of reason that "is cognitive and aesthetic, theoretical and practical, detached and passionate, subjective and objective".[62] For Augustine, this is an intellectual space formed, infused, and fulfilled within acts of love and praise. It is a rational space not ordered toward the uncovering of the self and God but, rather, toward the enfolding of the self into God. Here, Augustine offers us an inward space marked by the basic polarity of reason—mystery, wherein the law and rational organizing principle (*nomos*) of the self (*autos*) and its times is never its own self-transparent possession or creation. The space of the self is never, strictly speaking, private, autonomous, or value-neutral; it is a value-laden space that takes its existence and form through a rational-affective movement of openness (worship) toward God. Such openness is the only proper response for a being whose existence and identity finds its genuine selfhood precisely within the mystery of the infinite God: a being whose finite, mutable existence points to a self without absolute ontological foundation; a being whose identity as the finite image of God points to a self without an absolute existential foundation; and so a being whose nature as a *finite* being is constituted through what it essentially is not, namely, the infinite God. An account of the *saeculum* devoid of divine action—creative and soteriological—is inconceivable on Augustinian grounds in failing to understand this essential relation between God's power and creation and incoherent in failing to understand that the coherency of the soul—its rationality, morality, love—cannot take form without God.

This is also, then, not the inward temporality of the modern self that Gillespie details. The Augustinian self is not one whose interior space is constituted through a self-definition and self-possession of time—an autonomous self that constitutes "the times", the *saeculum*. This is not the *saeculum* of secularism, of a space and time controlled and defined through claims of human autonomy from divine power. Or, more accurately, it is an account of the *saeculum* in which such claims constitute the breakdown of human temporality: a sinful self-assertion of mutable creation claiming to define the times against the immutable creator with the inevitable result of the dissolution of the self and its proper temporality. This is precisely the dynamic Augustine confronts in the closing sections of *Confessions* 11 in the two routes he traces between being scattered in the times and being gathered in Christ.[63] Christ gives answer to the brokenness within the *saeculum*—a brokenness modern secularity does not rectify if the crisis of modernity articulated by its postmodern critics has any merit. In this, an Augustinian account of the *saeculum* contributes to more than simply the collapse of a supposed Medieval synthesis (*pace* Gillespie). Joseph Rivera charts one possible contemporary route in his argument that Augustine's social and political analysis of the two cities—the heavenly and the earthly—develops out of his anthropological appropriation of Paul's spirit–flesh framework.[64] Rivera argues that this framework allows for a different reading of the two cities than the standard route: rather than treating the cities as two distinct groups of people (i.e., the redeemed and the

---

[61] *trin.* 15.28.51.
[62] (Tillich 1951, p. 72).
[63] *Conf.* 11.29.39.
[64] (Rivera 2018, pp. 66–73).

damned), the two cities represent the soteriological and eschatological dynamic between spirit (grace) and flesh (sin) within each one of us. Rivera contends that this dynamic frames Augustine's account of the *saeculum*. I agree that soteriological and eschatological dimensions are central to Augustine's account and would add that both beckon toward Augustine's Christology and ecclesiology.[65] However, this final point will have to wait for further confirmation and detail in the following section as we move into the wider corporate context that shapes Augustine's account of selfhood and its reformation in Christ. Here, we will see that the inward space and time that constitutes the self is in significant part the consequence rather than the foundation of being already taken up within the times (*saeculum*) as they are reformed and reordered within the body of Christ.

## 3. The Times II: Human Community

There is continuity between Augustine's accounts of the *saeculum* as it is taken up within the human soul and wider human history. Contemporary commentators sometimes label these accounts his psychological/proto-phenomenological and his cosmological accounts of temporality. The former grounds time within the subjectivity of the soul, whereas the latter grounds it objectively in God's creative action. These two accounts are sometimes implicitly or explicitly assumed to be distinct or even in tension with one another. However, I would argue that a continuity exists between them that is grounded in Augustine's Christology. To see this, we need to return to his comparison between an individual, human history (the *saeculum*), and the recitation of a psalm, which occurs in the midst of his search to "hear the voice of praise and contemplate your delight".[66]

This search is anchored to the opening lines of *Confessions* 1 in Augustine's desire to find rest in God amidst the gap between mutable creation and the immutable creator—one rendered dangerous through sin.[67] Here, Augustine frames the *saeculum* within his doctrine of creation: the immutable God is "before the beginning of the ages (*ante primordia saeculorum*)" as the source of all mutable creation.[68] This mutable–immutable pairing provides the basic framework for the *saeculum* insofar as mutability is the foundation for the changes (temporality) that define the ages (the *saeculum*) but a mutability that exists through the creative power of the immutable God.[69] Augustine then ostensibly connects his account of the ages to his own age of infancy, highlighting the potential sinful actions in his early youth before rejecting any personal responsibility for them, because they are completely forgotten and no longer constitute his life now (*vivo in hoc saeculo*).[70] Augustine's response, however, belies an anxiety that drives his examination of the *saeculum*, namely, how the gap between mutable creation and the immutable creator has become exploited through the sin rooted within the *saeculum*. As such, Augustine's rejection of personal responsibility for the sins of his infancy carries with it the haunting recognition of a more fundamental sin that accompanies human life through the ages (*saeculum*).

Augustine's attempt to come to terms with this sin becomes apparent in, and indeed frames, his ensuing analysis of human language. We see this already in the opening lines of *Confessions* 1 in his movement between the eternal, immutable divine Word and temporal, mutable human words and his contention that our confession of faith and praise (human words) in Christ (divine Word) can overcome the sinful gap between God and humanity. This is possible because God enters time and history, bridging the gap in the incarnation to give us words to call to God,[71] which allows us to confess

---

[65]  Rivera and I agree that Augustine's theological anthropology, and the soteriological and eschatological dynamic that infuses it, frames his social and political analysis of the *saeculum*. In this, my own Christological and ecclesiological analysis of the *saeculum* complements Rivera's account. We do, however, disagree on certain aspects of Augustine's anthropology that center on the question of autonomy, with Rivera arguing for a more centralized place for human autonomy in Augustine's account than I think is present. In turn, this leads to different assessments of Augustine's legacy vis-à-vis liberal pluralism.

[66]  *Conf.* 11.29.39.
[67]  *Conf.* 1.1.1.
[68]  *Conf.* 1.6.9.
[69]  *Conf.* 12.6.6–12.7.7.
[70]  *Conf.* 1.7.11–1.7.12.
[71]  *Conf.* 1.5.5.

and praise the immutable and mysterious God.[72] It is this Christological and soteriological framework rather than one of infant psychology that structures Augustine's wider analysis of language in the opening books of *Confessions*. Here, Augustine is less interested in his own age than in the ages. That is, he is not after the natural origin and development of language within infants but, rather, after human language's origin and relation to divine language within this age (*saeculum*) in which mortal life (*vitam mortalem*) has become a living death (*mortem vitalem*).[73] This is the problem of language's origin tied into sin's origin read through Genesis 3:21, where Adam receives a coat of skins as a result of sin, which Augustine interprets as the reception of a mortal body. Augustine harbors deep suspicion against the way in which human language and culture function within this mortal age.[74] He compares the human customs that define the ages to a river (*flumen moris humani*), evoking imagery of human mutability as wild and chaotic, which carries "the sons of Eve into the great and fearful ocean which can be crossed with difficulty only by those who have embarked on the wood of the cross".[75] However, herein also lies the fundamental ambiguity of the *saeculum*, for it is an age defined through both the first and second Adam—through sin and grace. It is here that Markus moves too far in emphasizing the roots of the *saeculum* within the first Adam without adequately accounting for the place of the second Adam (Christ).[76] While human language and custom bear us from God into sinful mutability, the divine Word incarnate returns us to rest in divine immutability. Indeed, in *Confessions* 7, Augustine draws on the language of Genesis 3:21 to frame the incarnation: the immutable Word becomes incarnate not only in mutable existence but also in human mortality in order to open a route within the *saeculum* for human redemption.[77] This reminds us that the *saeculum* is not evil. Our salvation flows through the incarnation of the divine Word within human language and culture, and the driving force of *Confessions* is to find proper human words of praise and confession to unite us with the divine Word.[78] This unity reconnects us to the created origin of the ages[79]—ages that God forms through speaking the eternal, divine Word.[80]

Augustine's search for the proper words of confession and praise returns us once again to *Confessions* 11 and his employment of the Psalms in this endeavor. As we have seen, he concludes his account of temporality with a comparison between human life, history (the *saeculum*), and the recitation of a psalm. This is part of his attempt to understand how human words of praise, when voiced through the divine Word, elevate us to unity with God. This occurs through the incarnation, and so within the *saeculum*, in the place—namely, the Church—that God designates. We must now turn our attention to this place, focusing on the Christological hermeneutics that Augustine employs in his sermons on the Psalms. In particular, I would like to highlight Augustine's prosopological reading of the Psalms, which is an exegetical method that he utilizes to uncover hidden identities within the Psalms based on the context and content of the passage. He locates these identities within the shifting voices of the Psalms, focusing on four in particular: the twofold voice of Christ when he speaks in his divinity as the Word and in his humanity for sinful people; the voice of individual Christians as part of Christ's body; and the voice of the Church when it speaks corporately as the body of Christ. Through this exegetical

---

72　*Conf.* 1.4.4.
73　*Conf.* 1.6.7.
74　*Conf.* 1.13.20, 1.16.25–1.17.27, 2.6.14.
75　*Conf.* 1.16.25.
76　(Markus 1970, pp. 83, 94–95, 98–100).
77　*Conf.* 7.18.24. See also, *Gn. litt.* 11.31.40–32.42, where Augustine argues that Adam and Eve's realization of their nakedness signals that death and disease have entered the human body, which leads God to fashion coats of skin—mortal bodies—for Adam and Eve to replace the fig leaves in 3:7.
78　*Conf.* 5.14.24. One of the important ways Augustine comes to confess God properly is through the teaching and preaching (rhetoric) of Ambrose. In this, Ambrose stands as an important example and contrast to Greek mythology on the morally efficacious role of language.
79　*Conf.* 1.6.9.
80　*Gn. litt.* 1.2.4–1.2.6.

method Augustine develops important theological claims on the relation between Christ, the Church, and the individual Christian.

Michael Cameron has argued that Augustine's sermon on Psalm 21 is the linchpin to his reading of the early Psalms. Augustine's exegesis of it frames his wider Christological interpretation of the Psalms and highlights the way in which this interpretation takes shape through his developing Pauline theology of the cross.[81] Augustine begins his discussion of Psalm 21 by identifying its voice as the crucified Christ (*ex persona crucifixi*) speaking as sinful humanity (*personam … veteris hominis*).[82] He uses Christ's identity to puzzle over the question of what Christ means when he utters the words of abandonment on the cross. Because Christ is God incarnate and cannot be abandoned by God, it means that Christ is speaking for sinful humanity.[83] This symbolizes the sacrificial exchange that occurs on the cross between Christ and humanity: Christ takes on our sin and death, and we take on Christ's righteousness and eternal life.[84] In this, Augustine connects Christ's cry on the cross to the broader suffering that Christians undergo as part of their unity with Christ.[85] He argues that suffering is part of God's testing of Christians and the way in which God reforms sinful humanity.[86] Importantly, it is also the way that Christians come to embody Christ's passion and speak in Christ's voice.[87] Commenting on Psalm 21:5–6, Augustine argues that we are redeemed not only when we cry out to God in sin and suffering but also when we do so through the voice of Christ.[88] This, then, entails a certain "place" wherein our cry has soteriological efficacy, not within ourselves as individuals but within ourselves as we constitute the body of Christ; and, significantly, Augustine connects this voicing of Christ to following Christ's example, which entails an ethical and spiritual set of practices.[89]

Pertinent to our investigation is the social and political direction these sets of practices assume within Augustine's reflection on the body of Christ. To see this, we must first note the historical realism Augustine injects into his reflections on human suffering in Christ. A central passage here is Acts 9:4—*Saul, why are you persecuting me?* Augustine points out that Christ does not ask Paul why he is persecuting his saints but, rather, why Paul is persecuting Christ himself.[90] Christ claims this because the body (the Church) is joined to the head (Christ). Augustine interprets this joining through the dynamics of the incarnation. Drawing on the Christ hymn of Philippians, Augustine argues that Christ's downward movement into humanity begins the transforming of humans into Christ. In this exchange, the divine Word speaks in human words so that humans can eventually speak in Christ's voice.[91] Christ's protest against Paul in Acts reflects this exchange in voice and intimates a unity, grounded in love, in which Christ and the Church not only speak as one but also in some sense become one person.[92] Drawing on the marriage metaphor of Ephesians 5:31–32, Augustine maintains that the unity in voice between Christ and the Church reflects a deeper unity in flesh and personhood.[93] Christ and the Church are joined in the flesh of the resurrected human Christ,[94] which allows each to speak in the voice of the other.[95]

---

81   (Cameron 2012, p. 196).
82   En. Ps. 21.1.1.
83   En. Ps. 21.1.2.
84   En. Ps. 21.1.1, 21.2.3.
85   En. Ps. 21.2.5.
86   En. Ps. 21.2.5.
87   En. Ps. 21.2.8.
88   En. Ps. 21.1.6.
89   En. Ps. 21.1.7; Trin. 4.3.5–6, 4.12.15, 7.3.5, 8.5.7.
90   *En. Ps.* 26.2.11, 30.2.3, 32.2.2. Augustine sustains this line of interpretation into his later thought. For example, see: *ep. Jo.* 10.9.
91   *En. Ps.* 30.2.3.
92   *En. Ps.* 30.2.3.
93   *En. Ps.* 30.2.4.
94   *En. Ps.* 32.2.2.
95   *En. Ps.* 30.2.4.

While there is an eschatological trajectory to Augustine's discussion, given that we are dealing with the resurrected Christ and his mystical union with the Church, there is also a historical dimension in the way that Christ identifies with the current suffering of the Church.[96] This carries with it a distinctive social and political ethic (*pace* Markus).[97] This becomes apparent, for example, in the way that Augustine uses Matthew 25:35–45, where Christ claims that what one does to his followers is also done to himself, to develop an ethic around the shared identity between Christ and the Church. Augustine argues that the verses are not only evidence of the unity between Christ and the historical lives of Christians but also entail a duty of Christians to feed, cloth, and take care of one another.[98] This provides a basis and mandate for the Church's ethical action and involvement within the *saeculum*. Here, Augustine envisions the church as a social agent whose own normative standards govern its relations to wider society.[99]

This is not to say, however, that the eschatological body of Christ, as a heavenly vision of our future perfection, governs the ethics of the Church within the *saeculum* in a straightforward sense. That is, the heavenly ethics that guide the angels and saints do not govern the norms of the Church—the saint is not the norm of the *saeculum*. Here, we must push back against scholars such as Milbank who use the eschatological perfection of the city of God to delineate the normative standards for the historical Church, closely aligning the norms we are called to enact within the *saeculum* with those of the heavenly city. Admittedly, Milbank is right that the virtues are collectively enacted and that this points us toward the heavenly city as the source and unity for this collectivity. However, we should also heed Wetzel when he muses that Augustine's claim is even more audacious: the virtues are possessed only by God.[100] Perhaps there is an underlying connection between Milbank and Wetzel, namely, that the virtues are possessed by God and enacted collectively within the suffering body of Christ. This points us toward a model of virtue that works in and through the *saeculum* of human mortal existence as it unfolds within the ambiguity and mystery carried within the temporality of the soul. There is a play here, a movement, between the individual and the community, with Christ at the center, voicing mortal selfhood—the inner temporality of the soul in its mystery and ambiguity—and constituting the place—the corporate body, the Church—wherein this voice is elevated and perfected. In expressing human mortal selfhood in its ambiguity, we have seen that the central point of contact between divine and human words is the suffering Christ and the Church in its historical suffering. It is within this context that Augustine develops a model of virtue that seeks to align (train, habituate) the human voice of praise with the divine voice of creation and salvation. Here, Augustine brings forward a variety of moral activities—feeding the hungry, aiding the poor—firmly grounded in the ambiguity of the *saeculum*. More pointedly, for Augustine, the suffering in which the Church is involved may not simply be passive but may also entail coercion. Whether or not we would go this far with Augustine, the larger point is that the ethic that he draws from the body of Christ, which governs the Church, is rooted within the *saeculum* and its ambiguities—its sufferings and injustices.

Shifting metaphors from the city of God to the body of Christ provides occasion to revisit Markus' contention that Augustine inadvertently leads into a secular model of politics. Somewhat at the other end from Milbank, Markus argues that the ambiguity and mystery that accompany Augustine's use of the two cities metaphor lead unwittingly to a contemporary secular model of the *saeculum*: that is, a model of social and political space articulated in historical and political, but not theological, terms.[101]

---

[96] Thomas Breidenthal argues that, for Augustine, redeemed humanity comes to constitute the resurrected Christ's human form (Breidenthal 1998).

[97] (Markus 1970, pp. 83–84, 125–26).

[98] *En. Ps.* 29.2.22, 30.2.5, 32.2.2.

[99] This is in keeping with Raymond Canning, who argues that Augustine extends the love within Christ's body beyond Christians to the poor more generally (*minimi mei*), and with O'Donovan, who contends that Augustine universalizes the 'neighbor' (Canning 1993, pp. 383–94; O'Donovan 1980).

[100] (Wetzel 2004, pp. 282, 298).

[101] (Markus 1970, p. 104).

For Markus, the mystery of divine judgment over the *saeculum* means that we must withhold our own theological judgment on how political institutions fit, or do not, within the divine governance of the *saeculum*. Markus takes the eschatological distinction between the two cities, and our failure to have access to this distinction, as grounds for a political model. This unmoors political institutions from any definitive ethical or religious articulation, freeing such institutions from religion and leading, more or less, to a model of contemporary secular liberalism. Markus' argument has come under critique in various respects, which I will not revisit here except to note that one of the main issues that he misses is that the church/state model that emerges from Augustine's two cities framework relativizes politics so that it does not contribute to religion, but not vice versa.[102] Indeed, the political theology that Augustine develops is one in which the Church can and should be opportunistic in its engagement with political institutions, utilizing them for religious ends when available, all the while knowing that God and not the state is the final judge over the *saeculum*.

This conception of the Church as a social and political actor within the *saeculum* is grounded in Augustine's model of divine predestination. Bowlin is right here that despite Augustine's eschatological distinction between the two cities and the mystery of divine judgment that surrounds them, he spends a great deal of time articulating various ways in which the Church can and should be involved in politics, even in highly contentious issues, such as religious coercion. Markus would contend that Augustine is being inconsistent, contradicting the supposed separation between the church and state nascent within his two cities model.[103] However, if Markus' secularism thesis is right, then Augustine's inconsistency extends well beyond his politics. We have seen that Augustine's account of human selfhood is inconsistent and incoherent within a model of contemporary secularism. As such, if Markus is right, then Augustine's anthropology, and along with it his Christology and soteriology, becomes fundamentally inconsistent. Perhaps, however, the fault lies not with Augustine but with how we read his claims on the *saeculum* through contemporary social and political lenses. Here, again, I would suggest that we shift metaphors from the city of God to the body of Christ, underscoring that Augustine's recommendations on how the Church can and should engage the social and political spheres is, finally speaking, Christological. In particular, we need to note Augustine's claim in *On the Predestination of the Saints* that divine predestination grounds the incarnation and our inclusion within the body of Christ.[104] This signals, *pace* Markus, that the mystery of divine predestination is actually the basis for the Church's action in the world rather than a nascent foundation for secular skepticism.[105] Divine predestination grounds the incarnation, which, as we have seen, is also the definitive divine act within history that discloses the possibility of virtue within the *saeculum*. Divine predestination also grounds the Church, which acts within the political and social spheres as an *imitatio Christi* and, indeed, more profoundly as Christ, because the Church, in some sense, is Christ—the body of Christ. Here, we also see the connection in the way that the self and the Church embody the *saeculum* and how the Church, in some sense, is the foundation of the (reformed) self. Augustine's search in his soul for the proper praise and confession to articulate his identity as the divine image finds its soteriological foundation and fulfillment through participation within the body of Christ (the Church), which is the voice of Christ, who is the true and perfect image of God.

## 4. Taking Stock of the Times

Once again, this returns us to *Confessions* 11, because, finally speaking, the virtue that guides the Church within the *saeculum* is connected to the soul's own search for unity and wholeness in Christ. The Church's identity within the *saeculum*—the way in which it ethically occupies its social and political space—is connected with the space and time of the inward self and how it forms its identity.

---

[102] (Bowlin 1997, pp. 56, 59).
[103] (Markus 1970, pp. 144–46, 151–53).
[104] *praed. sanct.* 15.30–15.31.
[105] (Markus 1970, pp. 101–4).

Augustine's connection between individual life, human history, and the recitation of a psalm is the search for the voice that opens the way within the *saeculum* toward wholeness and peace. Augustine finds within the body of Christ this voice that gives power and order to his own voice of confession and praise and with it the hope for rest. We have seen this argument unfold in various steps, and here, I summarize the key claims. First, the *saeculum* emerges from finite, mutable creation as it forms from and is ordered by the infinite, immutable power of God. Here, humans have a unique role within creation according to the way in which the divine image forms and orders the inner nature, or space, of the soul in and through a direct relation with God. Second, within the *saeculum* humans encounter an array of ambiguity and hardship at various levels: at the level of virtue in immoral action; at the epistemic level in the failure to know oneself; and at the ontological level in the dissolution and division of the self. This ambiguity is rooted in a more fundamental mystery that enwraps human identity as a finite (created) image of the infinite (uncreated) God. This mystery becomes ambiguous—dangerous and treacherous—through sinful distortions in the human relation with God. Third, God enters the ambiguity of the *saeculum* in the incarnation of the Son in Christ to provide a perfect model (*exemplum*) for us to imitate that opens a path toward the reformation of the divine image within the soul. Fourth, this *imitatio Christi* comes in and through the Church, as the body of Christ, voices the divine Word within and through its own human words of confession and praise. This means that Augustine's search in *Confessions* 11 to overcome the dissolution of his soul within its sinful experience of the times—of human, mutable temporality unmoored from the immutable God—finds the "voice of praise" that leads to the "contemplation of delight" *within* the times of the *saeculum* as they are being reordered through the Church. Fifth, this points to a continuity, or order, between the self, the Church, and the *saeculum*—one grounded in the incarnation. There is a continuity—a call and response, if you will—between the inward space and time of the soul and the way in which the Church occupies the space and times of the *saeculum*. The soul's deformed temporality finds reformation as it finds its (divine) voice within the body of Christ. Sixth, there is an undeniable mystical and eschatological dimension to this process of unification, but it is grounded within the incarnation and so has a way of being that moves always through the historical times of the Church. Indeed, the unity between the Church and Christ appears most strongly within the ambiguity of the *saeculum*, namely, within the suffering of Christ and the Church. Here, the Church and Christ meet the soul within rather than above and outside the ambiguities of the times and the mystery of creation. This results in a set of spiritual and ethical practices that eschew both a heavenly ethic (Milbank) that reduces virtue to the practices of the saints and a secularist ethic (Markus) that fails to find an active and engaged voice of the Church within the *saeculum*.

**Funding:** This research received no external funding.

**Conflicts of Interest:** The author declares no conflict of interest.

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
