# Peer review of "An Augustinian Meditation on the Saeculum"

_religions, doi:10.3390/rel10080477_

Round 1

Reviewer 1 Report

Referee report on "An Augustinian Meditation on the Saeculum"

for Religions, open-access journal, 7/5/19

Rich, nuanced, well-informed, well-written and remarkably wide-ranging, this Augustinian "meditation" goes beyond mere exegesis of Augustine, but the middle path it takes between Milbank and Markus is, I think, faithful to Augustine's understanding of history assaeculum, an era of ambiguity that is neither sacred nor purely "secular" in the modern sense of the term.  It offers the refreshing realization that the "two cities" theme, which is radically eschatological, need not be the primary standpoint from which to examine Augustine's social and political thinking, and that we could take the the Body of Christ in time and history (i.e. in thesaeculum) as the primary standpoint instead.  In this regard, the connection drawn between the reading of a psalm as an image of the saeculumof history at the end of Confessions11 and Augustine's prosopological exegesis of Psalm 21(22), based on the unity of the resurrected Head and His suffering earthly Body, is particularly rewarding.  Likewise, the connection between the inner space of the self and the Body of Christ as the "place" of worship, where souls are united in love for God, is important - - though its liturgical dimension is developed here a bit further than I see it being developed in Augustine, who does not often dwell on the externalities of the church's worship.  Still, this further development is appropriate in a meditation that goes beyond mere exegesis, and it's possible to argue that Augustine could have developed this liturgical dimension further, and in much the same direction as this meditation, if he had had occasion to do so.  

This piece is clearly ready for publication as is, and is very likely to make a significant contribution to the ongoing discussion of Augustine on politics, history and the self. Perhaps at this point I might mention my most serious disagreement with the piece, as an initial contribution to the discussion about the piece. While I find myself in broad agreement with its treatment of both the temporality of the inner self and the spatial metaphors Augustine uses to describe it, I think there's an important omission, at least from the standpoint of exegesis of Augustine: the constitutive openness to worship that the author aptly identifies is grounded not on "mystery" but rather on the power of intellectual vision that structures the rational inner space of the self.  The "ambiguity" of this space in our fallen saeculumis best described, I think, in terms of the weakness and darkness of the mind's eye which call for a long process of healing and purification by faith.  To think in these terms, especially "vision" and "purification," puts the issue of Augustine's Platonist convictions squarely on the table for discussion, which I think is where it belongs.  

Author Response

I would like to thank the reviewer for the generous and insightful comments.  They were quite helpful.  I have made many of the changes that were suggested.  I would, however, like to leave the comments about mystery and ambiguity as they are.  The reviewer certainly has a point that for Augustine it is about clarity of intellectual vision and that he is following in a Platonist tradition.  I do not dispute this.  But with Augustine there are often multiple paths to tread on an issue.  For my article, I was trying to highlight the eschatological/ predestination angle, and I do think there is an element of mystery involved in this, though I am importing contemporary jargon to get at the issue.  Here I am following and implicitly responding to such scholars as: Dodaro, Barnes, and Wetzel.  And I would like to leave the language of mystery/ ambiguity to connect with such scholarship.  So, perhaps I can claim to be a bit more imaginative than exegetical at this point, with an eye toward contemporary scholarship.

Reviewer 2 Report

This is a superb essay. In the opening pages, the author sets his/her approach within the larger critical discussion of Augustine and the saeculum, knowing when to employ that discussion to support a refined reading of Augustine without getting lost in the weeds. In this way, the essay will resonate with readers deeply embedded in the scholarship but will not push away those who come to the essay without that knowledge. More important, the essay offers a new way of understanding Augustine on this topic (shifting away from the two cities metaphor to the body of Christ), and in doing so opens up news ways of understanding Augustine on key issues that run throughout his works (the self, time, knowledge, predestination, virtue, etc). Very jmpressive, and I have no substantial recommendations to make to this polished and persuasive essay.

Author Response

Thank you for the generous comments.